# Record, Recall, Reflect: A Qualitative Examination of Compassion Fatigue in Toronto Zoo Staff

**Krischanda Bemister-Bourret** [1,*] and **Merna Tawfik** [2]

1   Department of Psychology, Toronto Metropolitan University, Toronto, ON M5B 2K3, Canada
2   Department of Psychology, University of Guelph, Guelph, ON N1G 2W1, Canada; merna@uoguelph.ca
*   Correspondence: kbemiste@torontomu.ca

**Abstract:** This study explored Toronto Zoo animal-care professionals' (ACPs) experiences with compassion fatigue (CF) using a two-phase participatory methodology. In phase one, 11 participants took photographs of their workplace. In phase two, participants told the story behind their photographs through one-on-one interviews. The data were analyzed using NVivo12 software. The participants' experiences with compassion fatigue stemmed from issues with foundational infrastructure at the Toronto Zoo. Specifically, the participants highlighted issues related to training, staffing, and resource availability and discussed their resultant effect on animal welfare. The participants described the importance of built and sustained trust in their jobs, both with each other and with the non-human animals under their care. While the Zoo's motto is "One TZ", the participants noted conflict between the public's perception of the Toronto Zoo and how the organization cares for its staff. The additive effects of mental and physical exhaustion have led to disengagement from activities that once brought joy and difficulty staying focused while at work and home. The findings will enable the Toronto Zoo to provide comprehensive mental health support for their staff and allow participants, researchers, partner organizations, and the general public to discover more about compassion fatigue in the hope that the lessons learned will last a lifetime.

**Keywords:** animal care; compassion fatigue; human-animal interaction; Photovoice; qualitative research

## 1. Introduction

Those who work with animals in a professional capacity require a high degree of emotional and empathetic responsibility [1]. Animal-care professionals (ACPs) report high levels of work satisfaction [2,3] and are also at an increased risk for developing compassion fatigue (CF)—a state of "mental, emotional and physical exhaustion resulting from prolonged exposure to compassion stress" [4] (p. 253). CF is often referred to as the cost of caring [5], and the condition can occur from a single stressful event or continuous life situations that evoke feelings of frustration [6,7]. If such a state of being persists, the carer can experience feelings of isolation, sadness, mental and physical exhaustion, lack of concentration, compulsive behaviors, substance abuse, and increased interpersonal conflict [8]. As social isolation, helplessness, and difficulty dealing with work persist [1,8], the likelihood of depression, anxiety, and resentfulness increases [8–12]. The increased levels of cortisol from such stress may also impair cognitive abilities, leading to forgetfulness and impaired judgement, as well as psychosomatic symptoms such as migraines, headaches, vomiting, and nausea [8,13].

In a professional environment, CF can present itself as changes to emotional and organizational functioning (i.e., staff turnover and absenteeism, violations of company rules, changes in co-worker relationships, reluctance to change, and negativity toward management) [14]. ACPs with involvement in multiple caring roles tend to experience higher levels of CF than those with single roles [15], and workers who regularly engage in euthanizing animals experience increased job stress, work-to-family conflict, and physical

pain complaints [16]. Often, carers need to rely on their personal resilience to deal with traumatic events [17], but research suggests that organization-directed structural interventions may be more effective in supporting carers than wellness interventions aimed at individuals [18]. While an emphasis within animal care professions tends to be placed on physical safety (i.e., avoidance of bites, kicks, physical trauma, infections, sharp objects, etc.), there is a lack of training on mental health and how this can be impacted by working with and caring for animals [12].

Some recommendations have been put forth for developing resilience and coping tools for those who conduct research with animals. For instance, one program developed at the University of Washington called "Dare2Care" has begun work aimed at recognizing and raising awareness for CF, providing tools, strategies, and resources for emotion management for those working with laboratory animals. The Dare2Care program emphasizes both support for staff and the role of the workplace environment [12]. However, programs designed for reducing CF and research examining the effect of caring for animals on mental health have heavily relied on standardized measures and tools [1,12,15,19]; such an approach often misses the nuances gleaned from specific lived experiences and takes an approach of conducting research on participants rather than with participants. In contrast, Photovoice is a participatory method of data collection that allows participants to speak freely about their experiences and enables researchers to create knowledge that is closely centered on the individuals that their research aims to help [20]. Photovoice was initially inspired by Freire's (1970) theory of empowerment positing that power distributions between teacher and student as co-creators of knowledge are more effective than traditional teacher–student relationships [21] and feminist theory, which advocates for the active participation of individuals in the research process [22].

*The Site*

The Toronto Zoo is the largest zoo in all of Canada, spanning 710 acres in the Rouge River watershed and is divided into seven primary "zoogeographic regions", designed to represent the broad ecological divisions of the earth's animal populations: Africa (Rainforest and Savanna), the Americas, Australasia, the Canadian Domain, Eurasia Wilds, Indo-Malaya, and Tundra Trek (see Figure 1). The zoo first opened in August of 1974, and today, the organization employs over 273 permanent full-time employees and over 330 part-time or seasonal employees. There are also over 5000 animals currently on site, representing almost 500 different species [23].

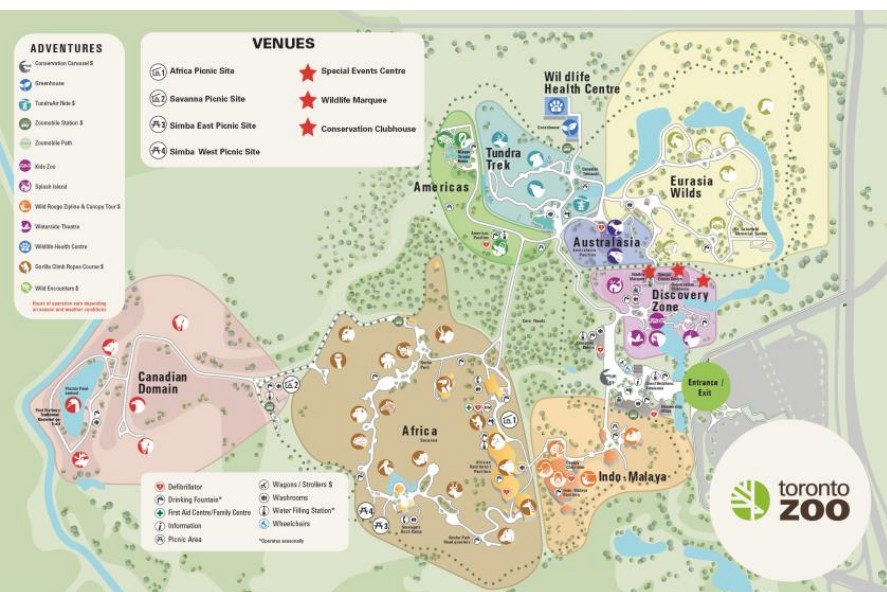

**Figure 1.** Toronto Zoo Visitor Site Map.

As an employer of a variety of ACPs (i.e., veterinarians, wildlife health technicians, keepers, reproductive research assistants, nutrition assistants, conservation stewards) [24], the Toronto Zoo provides a unique setting in which to study the impacts of CF on its staff using a participatory qualitative methodology. Animals in zoos are considered individuals and their quality of life is of high importance—as a result, many can live into very old age, and zoo workers often become attached to the animals in their care and grieve when they are ill, injured, or dying [25]. The complicated and multi-pronged demands of a zoo in a large and busy city such as Toronto likely puts these workers at a particularly high risk for CF, and the variety of perspectives of those employed at the Toronto Zoo provides a rich opportunity for investigation and intervention. The present study sought to understand CF in ACPs at the Toronto Zoo. No research to date has employed the use of Photovoice to explore the intersection of animal-care and mental health in a zoo setting.

## 2. Methods

The principal investigator (PI), Krischanda Bemister-Bourret, was responsible for funding and resource acquisition, investigation, conceptualization, methodology, software acquisition, data curation, supervision, and project administration; and jointly responsible for data validation, analysis, and writing of the present study. In late May 2022, the PI received an internal Toronto Zoo email address and sent an initial recruitment email to all wildlife care and wildlife health staff at the zoo. Three reminder emails were also sent throughout June 2022 to boost recruitment. In total, 11 employees at the Toronto Zoo (10 females, 1 male, aged 22–51 years) volunteered to participate in this study. The research received ethical approval from the Toronto Metropolitan University Research Ethics Board (REB#2022-177) and the protocol underwent peer review through Mitacs Accelerate Canada. The Toronto Zoo ACPs were contacted via email with an initial call for participants. To be eligible, the staff members were required to be at least 18 years of age; work at least 7 h per week on-site at the Toronto Zoo; have direct contact with animals in their job; and have cared for at least one geriatric, sick, injured, or dying animal. Participants received an honorarium of $40 for each phase of the project and identities were protected using participant numbers (001 to 011) and pseudonyms (Meerkat, Python, Cheetah, Hyena, Tamarin, Dragon, Otter, Kookaburra, Tiger, Wolf, and Catfish).

During phase one, each participant met with the PI and was provided with a verbal introduction to Photovoice. Within this meeting, the participants received one disposable camera each (with a maximum of 27 photos) and optional prompts to guide their photo-taking, such as, "Take photos that show what your job is like", "Take photos of things that concern you", "Take photos of things you'd like changed", "Take photos that show how you are feeling", "Take photos of things that you find physically or emotionally challenging to do". The participants were instructed to return cameras within one week of this initial meeting. Once returned, the PI digitally developed the photographs. The first participant began their Photovoice week on 13 June 2022, and the last participant completed their Photovoice week on 22 July 2022 ($M_{days}$ = 3.5, $M_{photos}$ = 16).

During phase two, the participants engaged in a 22 question semi-structured interview with the PI. Pseudonyms were used during the interviews and transcriptions. Aside from a list of general demographic questions, the participants were asked to describe their experience of photographing their workplace. The questions asked about participants' role(s) at the zoo, specific experience(s) with CF, supports in the workplace, recommendations for supports in the workplace, and personal coping mechanisms. The protocol questions were informed by the PHOTO technique [26] and Nicole Goettl's qualitative interview guide of CF among professionals working with survivors of sex trafficking (2016) [27]. The interviews were held at the participants' convenience on-site at the Toronto Zoo in a private and secure location and took between 60 and 210 min to complete. All interviews were audio recorded and were conducted and analyzed in parallel. There were a total of 545 pages of interview transcript. The first interview was held on 18 July 2022, and the last interview was held on 17 August 2022.

*Data Analysis*

The interview data were manually transcribed by a research assistant, and analyzed using NVivo 12 software, informed by a grounded theory method, to categorize segments of the participant responses. The grounded theory approach aims to generate theories from data where researchers systematically and iteratively analyze the data to identify emerging themes. NVivo12 is one of many pieces of computer-assisted qualitative data analysis software (CAQDAS) that do not analyze data but rather aid in the analysis process, allowing researchers to remain entirely in control [28]. The built-in capabilities of the program allowed for the creation of an extensive list of codes and subcodes, character-based coding, and rich text capabilities [28].

After an initial read through of all interviews, open coding began. The transcripts were read one line at a time with the goal of identifying salient themes in the data. Once all transcripts were open coded, the PI and research assistant focus coded the data. The common themes across participant interviews were identified and a working coding guide was developed. The transcripts were then read a second time, coded according to developed themes, and informed the creation of new themes, if necessary. Half of the data were coded by the PI and half by the research assistant, with an interrater reliability check completed at the conclusion. Once all interviews were focus coded, codes were further synthesized into three primary themes. Furthermore, select photographs and quotes were chosen to represent each theme following qualitative summary.

## 3. Results

A day in the life of an ACP at the Toronto Zoo consists of a wide range of activities and tasks. Some of the responsibilities discussed were directly related to animal care—such as feeding, diet preparation, cleaning, behavioral enrichment, socialization sessions, medication administration, and training. Other tasks were more administrative or "public facing" in nature, such as training animals to engage with the public, leading tours and "keeper talks", regularly checking emails, organizing staff and animal schedules, consulting with veterinarians, attending meetings, and liaising with plumbers and contractors. Three central themes emerged from interviews with participants. Specifically, "Issues with foundational infrastructure", "The importance of trust", and "The public vs. private Toronto Zoo" were most directly related to the participants' experiences with CF in their jobs. The participants noted several areas for improvement within the Toronto Zoo, all of which have implications for all organizations that employ ACPs.

### 3.1. Theme 1: Issues with Foundational Infrastructure

ACPs described the root of their experiences with CF as issues with foundational infrastructure. The primary issue noted within this theme was staffing and training. Participants noted that there is severe understaffing in animal care departments and that there are downstream consequences to such understaffing, such as feelings of burnout, pressure, and fear of not doing an adequate job in their role (see Figure 2). One participant stated, "The animals aren't getting proper care because they don't have enough staff" (TZ Staff Member, 2022). Lack of staffing also leads to physical and emotional exhaustion. The participants noted that many physical ailments have arisen because of their work, such as the need for elbow surgery, arthritis, back pain, and illness. Several participants noted instances when unexpectedly long days led to missing out on family time or interruptions in their personal lives.

When new staff are hired, the participants note that the process is competitive, which allows friction and negative energy to fester once team members are brought on. For instance, many seasonal staff come back to the zoo year after year and are not offered full-time jobs. Furthermore, the participants said that the "zoo struggles with diversity" (TZ Staff Members, 2022) and that much of this issue stems from the hiring practices currently in place. Furthermore, the participants do not feel properly equipped to train new staff, and they often wonder if they are doing a good enough job. The participants expressed

a need for more animal-care-related training to develop and maintain the specific skill sets needed for their roles. The participants state that the current training resources lack intentionality and are provided to "check the boxes" rather than with the goal to help staff grow. Furthermore, routines are not adequately structured or monitored. and the staff feel that the general work environment is "borderline unprofessional" (TZ Staff Member, 2022).

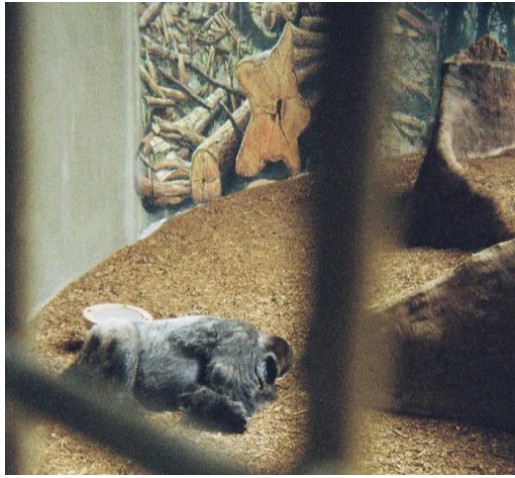
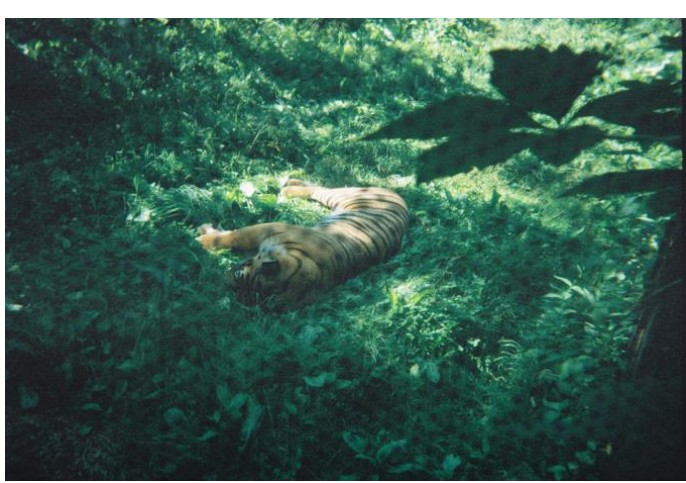

**Figure 2.** Sleeping gorilla (**left**) and sleeping tiger (**right**). Representations of physical and mental exhaustion. (Source: "Hyena" and "Python", TZ Staff Members, 2022).

The lack of appropriate resources is also a source of stress for ACPs. One participant noted that radios do not work adequately, which creates issues in daily communication. The participants also noted several occasions when issues related to animal welfare were not adequately addressed. For instance, epoxy (a chemical used in insulating materials, adhesives, and coatings [29]) has been a reported issue in the penguins' outdoor habitat since 2011, and the participants stated that there is a lack of maintenance of the penguin's air filtration system in the indoor holding (see Figure 3), which has led to a series of aspergillosis (a fungal infection that affects the lungs [30]) outbreaks. The participants also noted that rhinoceroses are housed on inappropriate concrete flooring and drink from a moat of water that is not properly filtered. One participant stated that the monkey house is "depressing", has "poor air quality", and is "falling apart" and that the warthog holding is a "gravel yard where they live alone" and is "completely inappropriate" (TZ Staff Member, 2022; see Figure 3). Another participant said that "half of the vents in the gorilla exhibit are broken", the tiger house is too small, and the orangutan exhibit resembles "a jail" (TZ Staff Member, 2022).

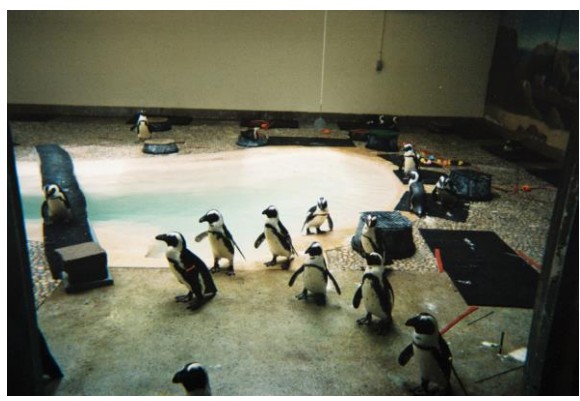
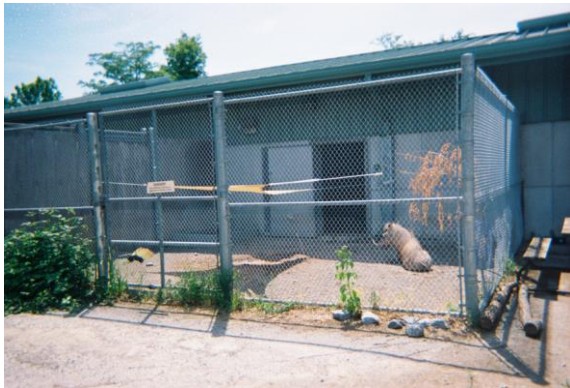

**Figure 3.** Indoor penguin holding (**left**) and outdoor warthog yard (**right**). (Source: "Tiger", TZ Staff Member, 2022).

### 3.2. Theme 2: The Importance of Trust

The participants described trust as fundamental to the work that they perform at the zoo, with one participant stating that it is a "mutual symbiotic relationship" that takes time to be earned and built (TZ Staff Member, 2022). When discussing trust that stems from communication with upper management, such as supervisors and managers, the participants noted that they often do not feel listened to by those in these positions. They have asked for help many times for issues related to animal welfare, but it went unnoticed and, eventually, they stopped asking. One participant said:

> When we're telling you that something is wrong, why aren't you listening? We're saying these things because we want there to be a difference. When we stop saying things, that's when you know there's a problem. (TZ Staff Member, 2022)

For instance, when staff have expressed feeling burnt out, needing a change in area, or needing a change in role, they have often not felt supported and sometimes even "threatened". As a result, there is a growing frustration around the decision-making processes at the zoo, and the "lack of animal related decisions being made by people [who work] with the animals" (TZ Staff Member, 2022). Participants do not feel that their opinion matters, and they do not have much faith in management when it comes to support for mental health challenges, in general, and CF, in particular. Furthermore, many participants noted that they do not feel that supervisors are "qualified enough" to deal with mental health challenges, and they avoid bringing such concerns to management. This is vital, as one participant said, "When you aren't coping with your emotions and when you aren't feeling things, you're pushing it down. But I also think you're holding back in future situations" (TZ Staff Member, 2022). Some staff feel segregated, gaslighted, and ignored when they have significant concerns to report (see Figure 4).

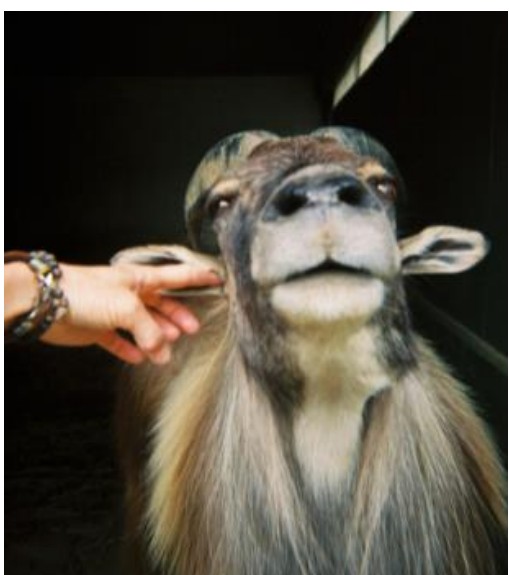

**Figure 4.** Representation of feeling unheard. (Source: "Python", TZ Staff Member, 2022).

Trust goes both ways in animal care professions. The participants also do not feel trusted by those in upper management—they do not feel that management values their skills or cares what "animal people" think. This is hurtful, and one participant said, "We know these aren't our animals. They're not ours. We don't own them, but we are their voice . . . we've dedicated our lives to taking care of these animals" (TZ Staff Member, 2022). The participants feel that management needs to trust staff more, consult with them on issues related to animal care, and follow through on what they promise. Moreover, the participants expressed a need to feel emotionally supported during difficult situations, such as animal death and euthanasia. One participant said:

> [We need to] intentionally build in conversations as a team ... not as coworkers and supervisors and employees, but as fellow humans ... we are all in this together. We can't ignore the human element of any organization, let alone an organization that relies so much on the care and compassion of their staff. (TZ Staff Member, 2022)

Furthermore, many participants commented on secrecy and deceit in the workplace. Communication was described as a "telephone game'", which adds pressure on ACPs and contributes to misinformation regarding animal care. One participant stated that there is a difference between "talking at people" and communicating with each other as a team. When discussing inter-staff communication, participants noted the importance and value of an "equal partnership" between the keepers and vets. This is a unique relationship, vital to the functioning of the zoo, and there was a plethora of positive comments related to the previous veterinary staff. During times when the vets had trusted and taken keepers' opinions into consideration, they felt incredibly supported. Some participants noted that the prior vet team went "above and beyond" to ensure that all staff had a chance to say their last goodbyes to animals before euthanasia and described them as an "air of calm" in stressful situations. With regards to the hiring of the new vet team, one participant said, "you can't just throw somebody in that position and expect [us] to trust them ... that's not how it works" (TZ Staff Member, 2022).

The participants' experiences, relationships, and bonds with animals at the Toronto Zoo were vital to their mental and emotional well-being. Many ACPs have developed deep, nurturing, and trusting relationships with the animals under their care. Participants described animals as their "best friend", "kind", "sweet", "lovely", "fighter", and "compassionate" and noted that moments of intimacy and understanding with the animals were what often helped them to cope with feelings of frustration and sadness—for many, working directly with the animals is their favorite part of working at the zoo (see Figure 5).

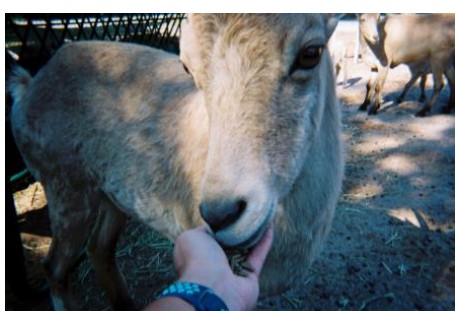 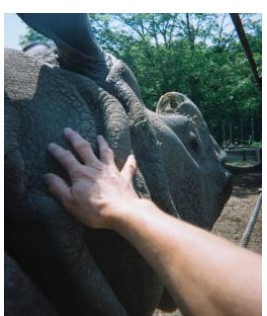 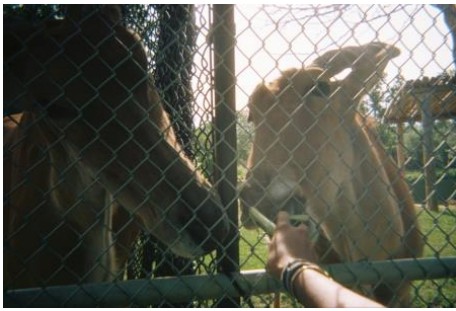

**Figure 5.** Participants noted that physically touching the animals is "therapeutic". (Source: "Kookaburra", "Python", and "Tamarin", TZ Staff Members, 2022).

The animals at the zoo are like family to staff; consequently, they can feel extreme frustration when their needs are not met. There are pros and cons to this type of attachment; some participants noted that they are "too attached" to the animals, but others say that this deep attachment is what drives their passion for their work, and they would not have it any other way. When animals pass away that staff have developed deep bonds with, it is "devastating". Some participants describe the experience like "losing a child" (see Figure 6). Many note that the pain is so overwhelming that it has led to "shutting off" emotions and "dissociating" in order to "guard [their] heart", cope with future loss, and to be able to continue doing their jobs (see Figure 7). One participant said:

> We're not supposed to look at these animals as ours. But they're ours, right? We spend more time with our animals than we do with our family. We know everything about them and we're a part of their world ... that's what we live for, is those connections. (TZ Staff Member, 2022)

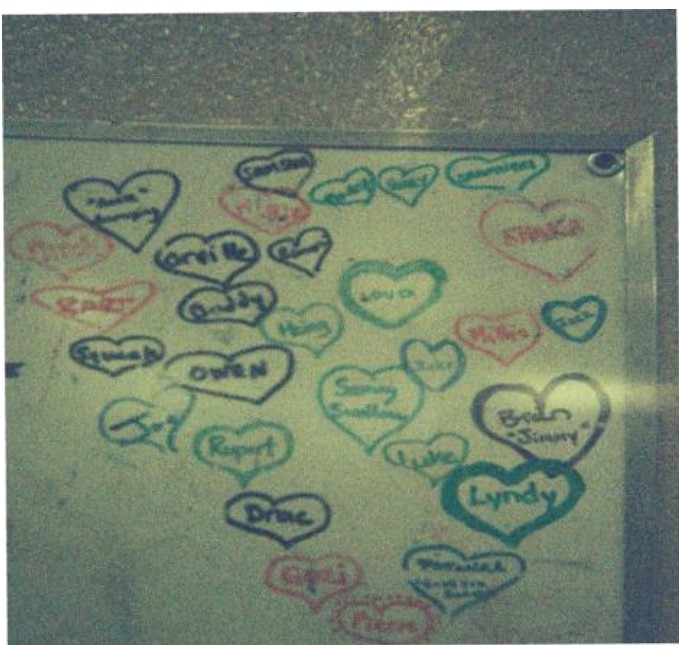

**Figure 6.** Memorial heart board in keeper room. (Source: "Wolf", TZ Staff Member, 2022).

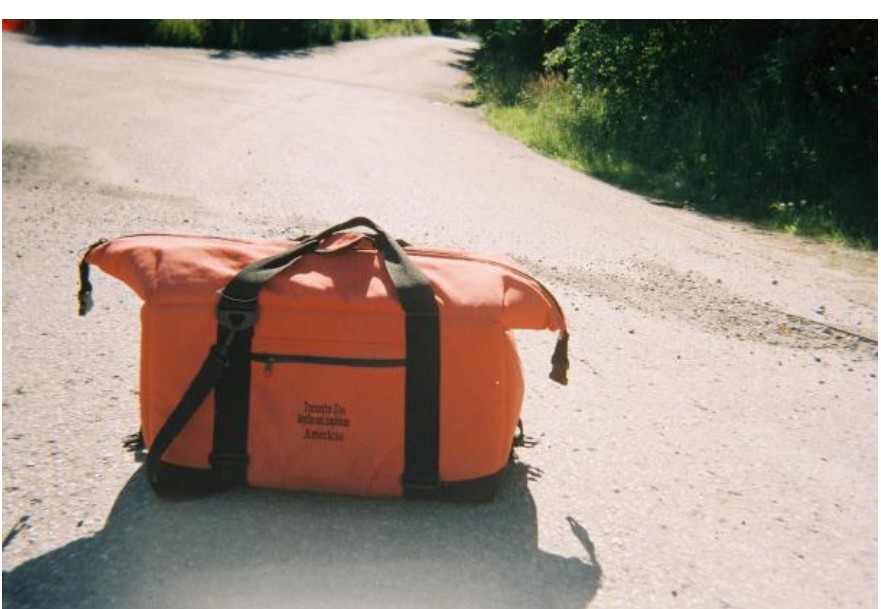

**Figure 7.** Toronto Zoo on-site medical bag. One participant said that this reminds them that "the majority of the times [I am] interacting with the animals, [they are] sick, injured, dead, or dying" (Source: "Dragon", TZ Staff Member, 2022).

*3.3. Theme 3: The Public vs. Private Toronto Zoo*

As a public organization, ACPs are often required to interact with visitors at the Toronto Zoo. Such interactions can take the form of official presentations such as keeper talks and Bush camp tours or informal interactions with visitors on-site. The participants discussed that such interactions are often positive experiences. One participant noted that when families are excited about an encounter with a particular animal during keeper talks, this "re-energizes" them and provides reassurance that they are impacting the public positively (see Figure 8). During moments like these, the participants felt that they were making a difference in people's lives.

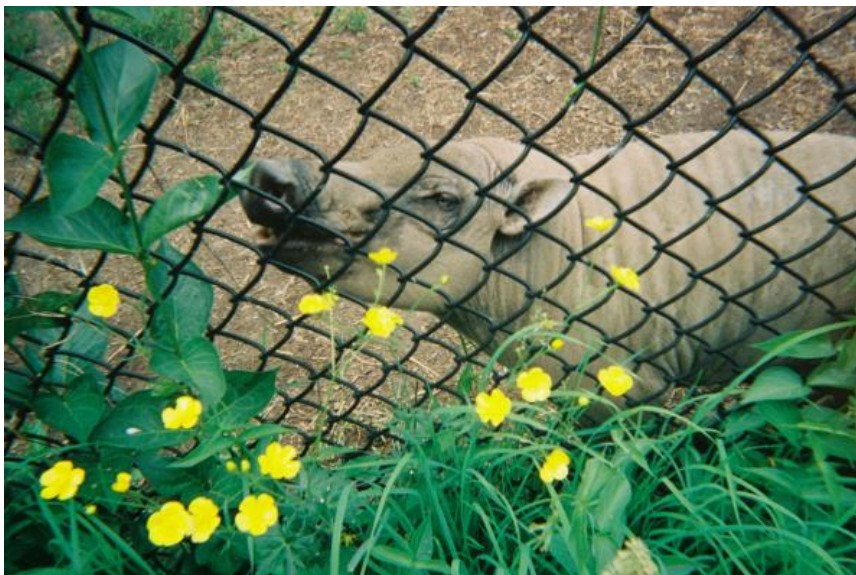

**Figure 8.** Babirusa amongst the flowers. One participant said: "it keeps you in the present moment to find things that you're grateful for, because even in the worst of days, there's still stuff that you can find that are positive and amazing" (Source: "Python", TZ Staff Member, 2022).

However, encounters with visitors were not always positive. The participants noted that they have sometimes felt that visitors interfere with animals' routines. The participants have felt that the Zoo values visitors more than its staff and that animal welfare is not prioritized. One participant noted that their "level of interaction with guests [can be] very rude" as a result, even though they feel there is a "constant pressure to keep people happy" (TZ Staff Member, 2022). Furthermore, the participants described feeling that the public has a "very false perception" of the zoo and its staff—visitors have made comments such as "this animal should be with its friend", "this animal needs more space", "other zoos are prettier", or "this animal should not be in captivity". Such interactions with guests can be "very difficult to deal with", "fatiguing", and "triggering", especially when staff have "virtually no power" to affect the type of change that guests comment on. One participant said:

> I think the Toronto Zoo is an amazing facility. I think we have lots of opportunities to do amazing things. I just think we need to get back to what we're here to do. And if we can't do that, then maybe we shouldn't be what we are. (TZ Staff Member, 2022)

One participant said that "80% of the sick time [they] have taken in [their] career has been for mental health days", and another said that they "hate who [they are]" at the Zoo. A third participant stated, "I'm looking back and I'm [wondering]—did I waste 25 years? [Would] I have had more of an impact doing something else or should I have just avoided trying to make a difference at all?" (TZ Staff Member, 2022). Many staff have noticed emotional numbness when looking at photos of animals that have passed—they find this disconcerting and wonder if it is because of emotional overwhelm and exhaustion. Some note that this sort of numbing is an intentional coping strategy, as "you can't feel every loss . . . it would just break you" and it is challenging to "just pick yourself up and go on" when feeling this way, particularly when something can happen at any time that is "retriggering or retraumatizing". For instance, some participants noted that watching the animals live in a "suboptimal state of welfare" where they are "not thriving" can lead to feelings of guilt and helplessness to re-emerge. The participants noted that while the Zoo's motto is 'One TZ', as well as 'We make change together', they feel that there is conflict around this messaging. The participants feel isolated, "in [their] own lanes" and that any change that happens at the zoo is not for the benefit of animals or staff. Whether it is

staff turnover, policy changes, changes to hours of operation, or accreditation procedures, nothing feels like it is done to "value every individual". One staff member said, "It's so easy to lose sight of the small gains that we are making because it feels like I'm just treading water all the time" (TZ Staff Member, 2022; see Figure 9). Staff constantly push for change within their area, but it is an "uphill battle" to see that change implemented. We make changes together? Staff say that "doesn't quite feel like truth" (see Figure 10).

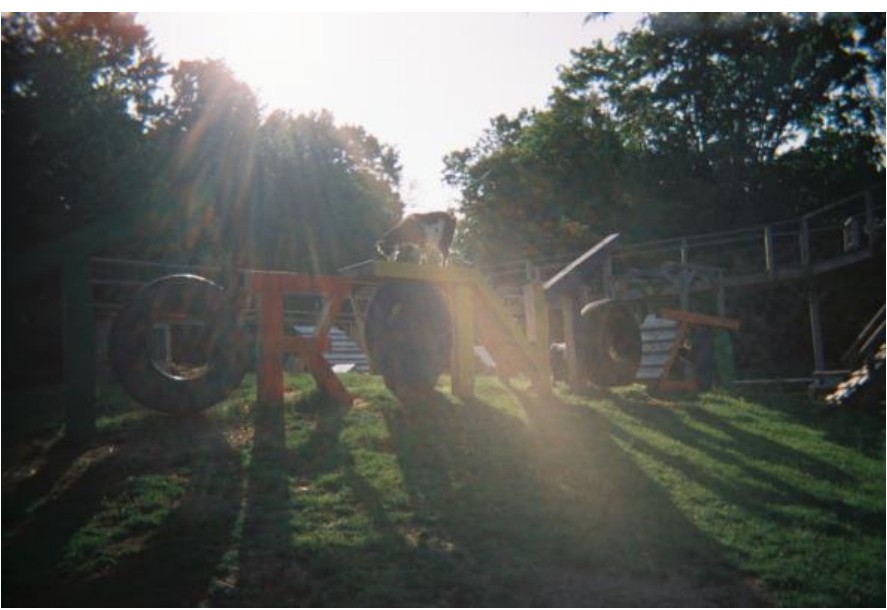

**Figure 9.** Goat atop Toronto Zoo sign. (Source: "Meerkat", TZ Staff Member, 2022).

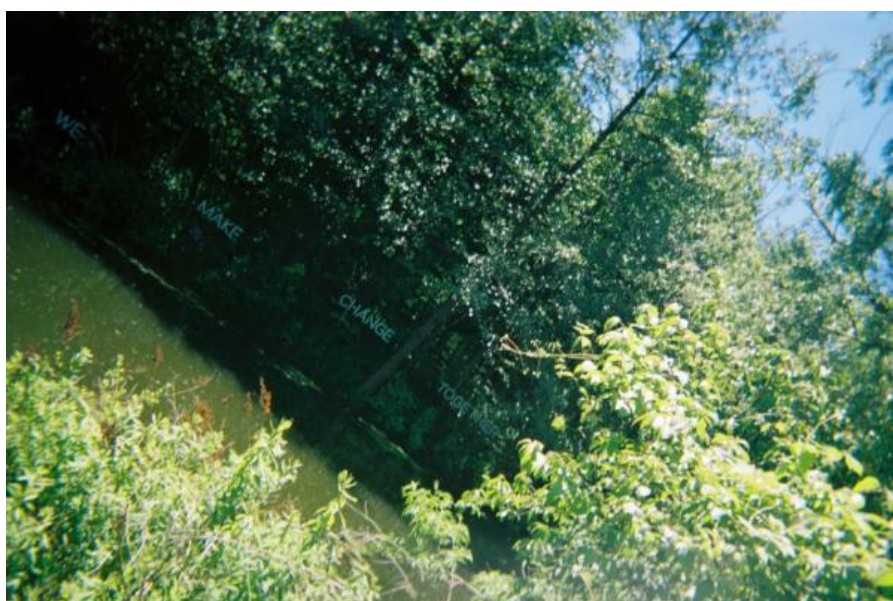

**Figure 10.** "We make change together" sign. One participant said: "There's a lot of mixed messages within the zoo culture. [There is a] real emphasis on positivity and working together towards a common goal . . . we're all in this. We're all supporting each other. [But it feels] like so often we're all just isolated . . . I want so badly to make changes to improve the lives of our animals and staff. And it just feels like a losing battle . . . so much change happens in this place and sometimes it just feels like it's not the kind of change that this sign is talking about" (Source: "Meerkat", TZ Staff Member, 2022).

## 4. Discussion and Conclusions

At the Toronto Zoo, ACPs care for a variety of animals and work in multiple areas, within a single day. Unlike zoos in the United States that may have keepers that specialize in a particular species (i.e., "big cat keepers", "hoof stock keepers"), keepers at the Toronto Zoo are 'generalist keepers' [31]. Generalist keepers perform daily tasks such as providing animal enrichment, cleaning exhibits, monitoring signs of injuries and illnesses in animals, assisting in the treatment and care of injured and sick animals, presenting educational talks for the public, and maintaining and repairing equipment and tools [32]. As a result of the demands of their many roles, ACPs have high expectations of themselves and internalize much of these expectations. This leads to feelings of overwhelm, and staff feel that they are not doing the best that they can for each individual animal within their care.

Consequently, staff are often unsure how to find the balance between advocating for animal welfare at the zoo and the barriers that they experience. The resources available are "limited and unsatisfactory"—ACPs fight, daily, to do the best for the animals in their care, but the participants note that they can only "do as much as the facility allows" (TZ Staff Member, 2022).

Challenges related to foundational infrastructure, trust, and communication at the Zoo, combined with the deep bonds that the participants in this study have with animals in their care, contribute to experiences of CF within this population. The participants frequently experience mental exhaustion within their daily jobs and the emotional toll from frequent traumatic experiences, such as animal illness, death, and injury, have long-lasting effects on their physical and mental health. Furthermore, the participants do not have enough time to research each individual species and fear that they are not doing enough for the animals in their care. This leads to feelings of guilt, inadequacy, and frustration. CF is often further exacerbated by factors such as poor work–life balance, "emotional labor", and excessive workload [33,34], and the participants in this study noted experiencing anxiety, stress, depression, and post-traumatic symptoms because of their work. The additive effects of this mental and physical exhaustion have led to disengagement for many ACPs. Staff are having difficulty staying focused while at work and home, and while they have a "desire to constantly learn and grow", sometimes they can only do the bare minimum. Staff also note feeling in a constant state of hypervigilance to ensure that they are noticing all signs of animal illness to prevent further death.

A note on COVID-19: While not directly addressed in this report, the COVID-19 pandemic was a significant source of stress in the lives of participants from 2020 to 2022. Stress stemming from general understaffing compounded during this time, and many participants noted how exhausting it could be to cover shifts due to unexpected illness. Furthermore, staff were concerned about the potential to pass COVID-19 to animals that were susceptible, and they had to wear personal protective equipment to protect these animals. This would often interfere with the bonding and enrichment programs that animals and staff were accustomed to. From the Summer of 2020 to the Spring of 2022, the teams within each area were also split in half to ensure that they were working on "A or B" rotations. Many staff noted that this process was "horrible", and one participant described it as "parents that are separated that have shared custody, but one parent doesn't talk to the other about what's going on" (TZ Staff Member, 2022). Everything took longer and was more difficult – as a result, the participants would often do the bare minimum of their job. One participant said, "we were just keeping these animals happy, keeping them fed . . . I just needed to make sure the animals were okay . . . and all the extra stuff just went away" (TZ Staff Member, 2022).

*Recommendations and Future Directions*

The participants in this study noted several areas for improvement within their workplace, many of which have implications for all organizations that employ animal care professionals. Staff would like to see intentional planning and goal setting within meetings, as well as trained conflict-management facilitators to promote a balance between bottom-

up and top-down management approaches. Supervisors should be trained on mental health first-aid, and staff would like the implementation of mandatory debriefs following traumatic animal encounters or deaths as a first step in countering the expectation that staff should just "move on as if nothing happened" or suppress their emotions following such incidents. The participants would also like to see the hiring of additional staff to support the increased workload demands of the zoo, as well as intentional investing in more training, such as zookeeper conferences and conservation training. Furthermore, there should be intentional efforts to increase diversity and fairness in the hiring process and more opportunities for non-work-related social activities between colleagues to increase trust within teams. Staff need to be listened to and consulted with when it comes to animal care or staff-related issues, and they would also like to see the distribution of educational resources related to mental health and compassion fatigue in ACPs. They would also like more extensive mental health benefits, as many staff noted that the therapy provided through the current Employee Assistance Program was not helpful to them. Staff would like more time to spend with animals to monitor the efficacy of enrichment and to ensure that they are appropriately prepared for events with the public. Finally, and very importantly, staff would like consistent and considerate recognition of the lives of animals that have passed.

While this study provided important and significant insight into ACP's experiences with CF at the Toronto Zoo, it is important to consider some important limitations of the research. Since participants were recruited on a volunteer basis, there is a predominant focus on White and female employees in this study. Furthermore, the sample size was limited, restricting the ability to generalize participants' experiences to the majority of ACPs that work at the zoo. One important contributor to the limited sample size was staff's concerns around confidentiality. Conducting interviews on-site, requiring participants to take photos with a disposable camera at their workplace, and the personal and intimate details of participants' disclosed experiences led to concerns by staff for management to assume their identifies. While immense care was taken to protect participant identities and present results in aggregate, participants were informed that confidentiality was not guaranteed due to the nature of the study. As such, many staff noted a fear of retribution from the zoo as a result of this work.

Further research is needed to investigate and understand CF in ACPs. The growing area of the human–animal bonds literature would benefit from an increasing focus on the relationships between ACPs and non-human animals. The research should also focus on identifying next steps in reducing barriers to promote human and non-human animal welfare [35].

**Author Contributions:** Conceptualization, K.B.-B.; Methodology, K.B.-B.; Software, K.B.-B.; Validation, K.B.-B. and M.T.; Formal analysis, K.B.-B. and M.T.; Investigation, K.B.-B.; Resources, K.B.-B.; Data curation, K.B.-B.; Writing—original draft preparation, K.B.-B. and M.T.; Writing—review and editing, K.B.-B. and M.T.; Visualization, K.B.-B.; Supervision, K.B.-B.; Project administration, K.B.-B.; Funding acquisition, K.B.-B. All authors have read and agreed to the published version of the manuscript.

**Funding:** This research was funded by Mitacs and Mental Health Research Canada (Mitacs Accelerate grant IT29137).

**Institutional Review Board Statement:** The study was conducted in accordance with the ethical requirements of Toronto Metropolitan University and approved by the Institutional Review Board of Toronto Metropolitan University (REB 2022-177, approved 25 May 2022) for studies involving humans.

**Informed Consent Statement:** Informed consent was obtained from all subjects involved in the study. Written informed consent has been obtained from the subjects to publish this paper.

**Data Availability Statement:** The data presented in this study are available on request from the corresponding author. The data are not publicly available due to privacy restrictions.

**Acknowledgments:** This work was supported by Mitacs and Mental Health Research Canada through the Mitacs Accelerate program. Authors express sincere appreciation to the Toronto Zoo organization and all participating staff. Authors also express gratitude to the host institution, Toronto Metropolitan University.

**Conflicts of Interest:** The funders approved the design of the study. The funders had no role in the collection, analyses, or interpretation of the data; in the writing of the manuscript; or in the decision to publish the results.

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
