# Peer review of "Record, Recall, Reflect: A Qualitative Examination of Compassion Fatigue in Toronto Zoo Staff"

_2673-5636, doi:10.3390/jzbg4020031_

Round 1

Reviewer 1 Report

The work is well written and novel. the authors could extend and improve the discussion; however, the literature on the subject is scarce.

Author Response

Thank you very much for your review and comments. Some paragraphs previously in the results section have been shifted to the discussion, for the purposes of both reducing the amount of detail in the results section and extending the discussion section. Further, a limitation and recommendations section has been added to the discussion.

Reviewer 2 Report

Limitations presented by the low sample size, focus on female zoo employees, and the likelihood of retribution from the zoo as a result of this work (photos were taken, etc. leaving little guess at who reported what) should be addressed/acknowledged.

Author Response

Thank you very much for your review and comments. A limitation and recommendations section has been added to the discussion.

Reviewer 3 Report

 Record, Recall, Reflect: A Qualitative Examination of Compassion Fatigue in Toronto Zoo Staff

The authors explored the Toronto Zoo Animal-Care Professionals’ experiences on Compassion Fatigue, using a participatory methodology concept and the collected data were analysed using NVivo12 software. Such study is essential for improving the animal welfare and zoo management and thus ultimately enhancing the conservation standard. However, there are some major suggestions especially regarding results section, discussion and need for new section on recommendation. In addition, there are a few small suggestions, and all these are listed one-by-one below. If all these are incorporated into the revised version, the manuscript would become clear and comprehensive.   

Comments and suggestions.

Line No. 7-8: This study explored Toronto Zoo animal-care professionals’ (ACPs) experiences with 7 compassion fatigue (CF), using a participatory methodology.

Suggestion: In the above sentence introducing the two-phase concept, will give better flow with subsequent statements.

Line No. 12-13:  Specifically, participants highlighted issues related to training, staffing, and resource availability, and discussed impacts on animal care. 

In the above sentence, I hope what the authors mean by ‘discussed impacts of animal care’, is the consequences of issues related to training, staffing and resource availability on animal care. And if so, the second part of the sentence i.e., ‘and discussed impacts on animal care’ may be modified as either ‘and their consequences on animal care’ or ‘and their resultant effect on animal welfare’ could give better effect and easy understanding to the readers.

Line No. 22-23: Are the words keys to be arranged in alphabetical order?

Line No. 26:  Those that work………. emotional and empathetic responsibility.

In the above instead of the word ‘that’ ‘who’ may give better flow.

Line No. 101: Participants received an honorarium of $40

In the above use of word ‘an’ is improper. Replace this ‘a’

Line No. 105: During phase one, each participant met with the principal investigator (PI) and were provided with a verbal introduction to Photovoice.

In the above, the Principal Investigator needs to be introduced. Who was the PI and what was his responsibility in the study.

Line No. 110-111: Participants were given one week to take photographs of their workplace environment from the day of the initial Phase one meeting, taking notes to pair with photographs at their discretion.

The above statement is not clear. Please rewrite it.

Method

The section is very clear but the Line No. 94:  11 employees at the Toronto Zoo (10 females, one male, aged 22-51 years) participated in this study.

The authors mentioned of the total ACPs, 11 participants were sampled with 10 of them women and one man. I suggest on what basis sample size 11 were decided and similarly gender selection why it is biased towards women? Need to be explained.

Excepting for the above one lacuna, rest of the method section are quite clear.

Results

The result part of is too descriptive. Although topic of the manuscript concerned needs elaborate presentation of the results, it is too elaborate.

I suggest the 11 participants opinion could be segregated into issues like foundational infrastructure, trust, and communication and how that would affect directly or indirectly affect the welfare of the participants and welfare of the zoo animals could be produced as a table, so that majority of the participants opined towards which category and what kinds of consequences each of will leads to, if zoo management does not care about could also be shown within the same table. Such table could make the authors to reduce the text drastically. And the such detailed table also easier to zoo management concerned to visualize its lacunae.

Discussion

This section is precise, and I feel the consequences of mismanagement of ACPs by zoo component described in the result section could be moved to discussion.   

For such manuscript I further suggest the authors should include an exclusive section for recommendation. Ultimately such study is aimed at improving the standard of Zoo set-up, a recommendation section is a must, even if journal format does not have this section but for such paper the section is essential.            

Author Response

Thank you very much for your detailed and thorough review and comments.

Your suggested minor word changes and additions have been made to the abstract, introduction, and methods section to increase clarity.

Larger changes have been made to the results and discussion sections. Some paragraphs previously in the results section have been shifted to the discussion, for the purposes of both reducing the amount of detail in the results section and extending the discussion section. Further, a limitation and recommendations section has been added to the discussion.

You recommended that results could be produced as a table, in order to reduce text and enable zoo management to easier visualize the results. While we understand and appreciate this suggestion, we feel that the current presentation of result summaries, interspersed with photos and quotes, presents the results in an effective manner for this particular manuscript. The Toronto Zoo has been provided with shorter, more visually appealing knowledge mobilization handouts of results.